# Stretchable and High-performance Sensor films Based on Nanocomposite of Polypyrrole/SWCNT/Silver Nanowire

**DOI:** 10.3390/nano10040696

**Published:** 2020-04-07

**Authors:** Bu-Yeon Hwang, Wen Xuan Du, Hee-Jae Lee, Sungmin Kang, Masaki Takada, Jin-Yeol Kim

**Affiliations:** 1School of Advanced Materials Engineering, Kookmin University, Seoul 136-702, Korea; 01087896432@kookmin.ac.kr (B.-Y.H.); duwenxuan1314@naver.com (W.X.D.); crocap93@kookmin.ac.kr (H.-J.L.); 2Advanced Technology Research Dep. LG Japan Lab Inc., 4-13-14, Higashi Shinagawa, Shinagawa-ku, Tokyo 140-0002, Japan; sungmin1.kang@lgjlab.com (S.K.); masaki.takada@lgjlab.com (M.T.); 3Institute of Innovative Research Laboratory for Future Interdisciplinary Research, Tokyo Institute of Technology, Yokohama 226-8503, Japan

**Keywords:** silver nanowire, single-walled carbon nanotubes, polypyrrole, stretchable, transparent, gas sensor

## Abstract

We report the fabrication of stretchable sensor films (SSF) using a composite of functionalized polypyrrole- single-walled carbon nanotube (SWCNT)-silver nanowire hybrid networks embedded into a cross-linked polydimethylsiloxane elastomer. The SSF exhibited low resistivity of 30 Ω/sq and an outstanding mechanical elasticity of up to 25% (no visible change in the sheet resistance after 100 cycle at stretching-release test of 25%). These SSFs were responsive to 1 ppm ammonia gas even at a low temperature of 40 °C with 20% relative humidity and also maintained reproducibility and reversibility when repeatedly exposed to ammonia gas more than 100 times. In addition, it was confirmed that the sensor film was hardly affected even at a relative humidity range of 20% to 80%.

## 1. Introduction

Stretchable electrode films (SEF) have increasingly attracted attention owing to their potential uses for many devices such as flexible displays, sensors, actuators and energy storage devices [1,2,3,4]. In particular, these SEFs, which have recently begun to be widely used as sensor-films, respond to mechanical deformations by the changes in electrical characteristics, such as resistance, owing to their stretchability and reproducibility. Nowadays, SEF applications, as a gas sensing film that responds to environmental stimuli (i.e., chemical, electric, magnetic, piezo stimuli) to produce a dynamic and reversible change in their properties [5], have attracted considerable interest owing to their promising applicability to sensors [6,7]. In particular, these stretchable sensor films (SSF) have great significance in being applicable to various wearable electronics such as soft robotics [8], artificial e-skins [9], human skin-attached film sensors, portable chem-resistive gas sensors and health care monitoring/diagnostic devices [10]. To make skin-mountable devices or wearable, stretchable, defined by the failure strain, is essential as well as flexibility. For this purpose, several types of stretchable sensors have been developed through the combination of nanomaterials with excellent electrical and mechanical properties and elastic polymers [11,12,13]. However, flexible and stretchable sensors of thin film type have become increasingly attractive due to the number of advantages such as miniaturization and portable or wearable properties.

In this regard, nanomaterials such as silver nanowires (Ag NWs) [4], single-walled carbon nanotubes (SWCNTs) [14,15,16] and graphene sheets [17,18,19] and their composite structures, have been reported to be widely used as sensitive strain sensors, which have an electrical conductor [20,21,22,23]. Recently, strain sensors based on a nanocomposite of an Ag NWs network embedded between two layers of a polydimethylsiloxane (PDMS) elastomer has been reported to exhibit high stretchability up to 70% with strong piezoresistivity [4]. In addition, conducting polymers such as polypyrrole and polyaniline and their metal oxide nanocomposites have been reported to demonstrate considerable potential as an efficient chemical gas [24,25,26,27,28] or pH-responding materials [29,30] owing to their unique electrical, optical and mechanical transduction mechanism [31,32]. Our group [27] has also reported that the sensor films comprising polypyrrole nanoparticles (PPy NPs) complexed with phenylalanine (PA) sensitively react to ammonia gas and exhibit significantly improved reproducibility and reversibility upon gas exposure. Fu et al. [30] have been reported an electrical switch for smart pH sensor films based on Ag NW-polyaniline nanocomposite. However, despite of the several studies being conducted on transparency, extensibility and sensitivity, smart sensing materials or structures still do not completely satisfy the requirement as sensor with controllable chemicals, pH, piezo stimuli and rapid response at a low voltage.

Herein, we report the highly sensitive, selective, extensible and reliable “smart SSF” based on a composite of functionalized PPy NPs with SWCNT–Ag NW hybrid networks embedded into cross-linked PDMS elastomer, as shown in Figure 1. In particular, the hybrid conductor comprising an Ag NW network interconnected with SWCNTs embedded into PDMS exhibited high elasticity, cycling stability and excellent sensitive at low voltage. In addition, the SWCNT–Ag NW hybrid network structured conductor films were confirmed to exhibit good response to the stretch/release for ≥100 cycles and hysteresis tests without the loss of conductivity under stretching conditions of 25% were also conducted. PA-complexed PPy NPs as effective chemical sensing materials were also prepared by incorporating PA into the conductive PPy backbone, which was hybridized with the SWCNT–Ag NW hybrid networked conductor. The PPy-PA complex NPs exhibited effective reproducibility and reversibility when exposed to chemicals such as ammonia, which were characterized by the ability to increase the sensitivity by 2-fold to chemicals via hybridization with SWCNT-Ag NW hybrid networked conductors.

## 2. Experimental Section

### 2.1. Preparation of Materials

SWCNT powder (diameter 1–3 nm, length 3–5 μm) purchased from Hanwha Nanotech Inc. was used as received. Hydrothermal treatment was performed using concentrated HNO_3_:H_2_SO_4_ (v:v = 1:3) following the procedure described in our previous paper [33]. Acid-treated SWCNTs functionalized with carboxyl groups (80 mg) were added to 50 mL of deionized (DI) water with polyvinyl-pyrrolidone (PVP; average molecular weight M_w_ = 4000; Sigma Aldrich, St. Louis, MO, USA) as the dispersant. After an ultrasonic dispersion treatment for 3 h, centrifugation was performed at a speed of 12,000 rpm for 30 min. Finally, the supernatant was collected to obtain an acid-treated pure SWCNT mono-disperse solution. Ag NWs with an average diameter of 20 nm and a length of 20 μm were directly synthesized according to the magnetic-ionic-liquid-assisted polyol process [34]. As-synthesized Ag NW was re-dispersed in 0.5 wt% DI water and prepared in an aqueous suspension solution. The conductive Ag NW-SWCNT hybrid aqueous ink solutions were prepared by a direct mixing method of the aqueous suspension of 20-nm-thick Ag NWs and PVP-dispersed SWCNT aqueous solutions. 

Functionalized conductive PPy particles coupled with phenylalanine (PA), as an ammonia sensing material, were prepared according to our previous paper [27]. First, PVP (average molecular weight 1,300,000) was dissolved in water. Second, a 0.1 M Fe(III) Cl_3_·6H_2_O solution (Sigma, 98%) was added to 1 wt% PVP solution as the oxidant and the temperature was maintained constant at 0 °C. Next, 0.03 M PA and 0.043 M pyrrole monomer were added in the order specified. The mixtures were stirred during polymerization for 10 h, affording spherical PPy–PA-complexed nanoparticles (NPs) of 40~60 nm size. Subsequently, the PPy powder washed several times with methanol and then dried in a convection oven under vacuum for 12 h at room temperature. Finally, the conductive PPy–PA-complexed NPs were dispersed in DI water.

### 2.2. Fabrication of the SSF 

SSF were fabricated as following; First, Ag NW-SWCNT hybrid aqueous ink solutions were spin-coated on a Si wafer, which was previously cleaned with acetone, following by drying at 80 °C for 5 min. Second, 0.01 wt% silica gel dispersed in ethanol was spin-coated at 1000 rpm and post drying, liquid PDMS with a thickness of ~50 µm was coated on the upper surface of silica and the Ag NW–SWCNT hybrid layer, followed by curing and crosslinking at 180 °C for 30 min. Third, after peeling the cured PDMS from the Si wafer, it was flipped and copper wires were attached to the ends of the embedded Ag NW–SWCNT hybrid layer using a silver paste for further mechanical and electrical tests. Finally, the PPy–PA complex NP aqueous solution was coated on the upper surface of the Ag NW–SWCNT hybrid layer and the finished SSF samples were dried in a vacuum chamber of 80 °C for 30 min. However, the fabrication process of SSF based on a nanocomposite of PPy-PA-complexed NPs with SWCNT–Ag NW hybrid networks embedded into the cross-linked PDMS elastomer is summarized in Figure 2.

### 2.3. Sensor Characterization and Detection

The gas sensors thus prepared were set in a test cell to measure the resistance using the electrochemical detection technique. To investigate how the sensitivity of SSF depended upon the NH_3_ vapor concentration, each sensor film (sample size = 30 × 30 mm) was placed in a 350-mL vessel maintained at 100 Torr into which dry air as a carrier gas and then NH_3_ gas at varying conditions (1–5 ppm) were introduced. The current was measured in real time using a digital multi-meter (Keithley 2000) under an applied dc voltage of 2 V at 40 °C. Dry air was passed through the vessel until a steady current reading had been obtained. The 0.033 mL aqueous solution of NH_3_ (1 ppm) was then introduced in the vessel for 30 s with dry air as a carrier gas and the current was recorded. The 1 ppm NH_3_ (aq) gas flow was turned off and the chamber was left to recover under dry argon gas. This process was performed repeatedly several times. During this process, argon gas was passed through the chamber until a steady current was reached. The NH_3_ response properties were carried out at 40 °C condition and the interference of humidity was also examined at 20, 50 and 80% RH (Relative Humidity) conditions, respectively. In particular, in order to stably maintain the 20–80% RH conditions, the temperature of the chamber was controlled at 40 °C. Test chambers for controlling relative humidity were self-made. The test cells were placed in a humidity chamber. The humidifying gas was prepared by mixing a dry gas and a gas generated by bubbling pure water and gas flowing it into a chamber in which a teat sample was installed. The humidity was adjusted by controlling the flow rate ratio between the dry gas and the gas bubbling pure water. The flow rate of each gas was controlled using a mass flow controller and the humidity measurement was monitored using a humidity sensor installed in the gas flow path. 

## 3. Results and Discussion

Figure 1-(I) and Figure 3-(I) show the cross section of a sample of the fabricated SSF and their SEM image and material structures prepared using a composite of functionalized PPy as the sensing material with high-conducting SWCNT–Ag NW hybrid networks embedded into the PDMS elastomer and Figure 1-(II) show a SEM image of the surface of the SEF. As shown in the model description in Figure 1, PPy–PA-complexed NPs of functionalized PPy as the sensing material, comprising PA-bound PPy NPs, have been reported to be potentially applicable as good chemical sensors [27]. Basically, they are connected via a layer-to-layer hybrid to the SWCNT–Ag NW hybrid network conductors. In this study, PPy–PA-complexed NPs were prepared via emulsion polymerization using PVP as the capping reagent. Spheroid-type well-defined PPy–PA-complexed NPs were obtained without aggregation (Figure 3-(II)). Changes in reaction conditions such as PVP concentration and temperature did not affect the shape of the resulting polymer; however, the NP size was strongly affected by the PVP molecular weight. Figure 3-II-(a to c) shows the structures of PPy–PA-complexed NPs obtained using different PVP molecular weights of 40000, 360,000 and 1300,000, respectively. Using a PVP molecular weight of 40,000, NPs with sizes ranging from 80 to 110 nm were obtained, whereas on using PVP molecular weights of 360,000 and 1300,000, NPs with sizes ranging from 60 to 90 nm and 40 to 60 nm were obtained, respectively. PA was incorporated into the PPy backbone via charge–charge interactions between the positively charged polaron of doped PPy and the appropriate counter ion of PA. PPy–PA NPs were found to be electrically good conducting materials, with a conductivity of ~12 S/cm and could be used for redox reactions. In addition, the doping level of PPy–PA NPs, which can determine the degree of conductivity, can be changed via oxidation or reduction, leading to the change in conductivity. With this characteristic, PPy–PA NPs can exhibit sensitivity to specific chemical species following the redox process, for example, ammonia and these NPs can demonstrate excellent sensitivity via enhanced interaction between the PPy–PA NPs and analytes. In particular, when a functional structural material (here, PA) with a carboxylic group and an amino group symmetric to each other, as shown in the chemical structure of Figure 3, was electrically coupled to the electro-conductive PPy main chain, the positive charges of an amine group (NH_3_^+^ region of PA) were sensitive to NH_3_ [27]. In this process, the lone pair of NH_3_ selectively reacts only with the NH_3_^+^ part of PA without affecting the positively charged (polaron) structure comprising the main chain of PPy. At this instance, the current density of the main chain of PPy increases with the flow of the external current. Conversely, after the removal of NH_3_, the bond between NH_3_ and NH_3_^+^ is broken, returning to the current at the existing state. This reversible reaction is a structural feature of a PPy–PA-complexed polymer having a π-conjugated double bond system as the sensing material. However, based on the abovementioned reaction mechanism, PPy–PA NPs can react via rapid adsorption and desorption kinetics for analytes, leading to a rapid response–recovery time.

Figure 4-(I) shows a photograph of the finally produced stretchable sensor film sample and Figure 4-(II), (III) and (IV) shows the film’s SEM surface images of approximately 50-nm PPy–PA-complexed NPs of 1, 2 and 3 wt% coated on the Ag NW–SWCNT hybrid conductive layer with a sheet resistance of 30 Ω/sq, respectively. PPy–PA-complexed NPs with diameters of 40–60 nm was uniformly dispersed on the top of the Ag NW-SWCNT network layer. Next, PPy–PA-complexed NPs were interconnected with the Ag NW–SWCNT network, affording a junction of possible mutual electron transitions and the Ag NW–SWCNT network improved the electrical conductivity owing to the electrical force and also acted on the electrical spreaders in the sensor films. This ensured good contact between the Ag NW–SWCNT network and PPy–PA-complexed NPs, leading to an advanced electrical property of the conductor. In particular, as a result of coating 1, 2 and 3 wt% PPy-PA-complexed NP solution on the Ag NW–SWCNT hybrid conductor, the sheet resistance of the conductor film itself was 30 Ω/sq, regardless of the concentration of PPy-PA-complexed NP. However, the sheet resistance did not change with the Ag NW–SWCNT network conductor before the PPy–PA-complexed NP coating but different light transmittances were observed depending on the density or concentration of the PPy–PA-complexed NPs. Herein, a highly sensitive, extensible and reliable “smart SSF” based on a bilayer hybrid structure of PPY–PA NPs and an Ag NW–SWCNT hybrid layer embedded in the surface layer of the cross-linked PDMS elastomer film, with characteristic elastic deformation is reported. PDMS completely penetrated into the Ag NW– SWCNT network and filled the gaps between Ag NWs and SWCNTs, affording an Ag NW–SWCNT hybrid network and PDMS. 

Figure 5 shows the elastic behavior for the sample under dynamic loading. In Figure 5 (I), the change in R/R_0_ at strain restraint for tensile strain at 15%, 20% and 25% tensile strain was observed at 25 °C room temperature. The initial sheet resistance (R_0_) was almost completely recovered for a stretch/release cycle test with strains ε between 15% and 25%, revealing the outstanding stretchable property of film sensors (Figure 5 (I)). Given that the flexible and stretchable characteristics of PPY–PA NPs/Ag NW–SWCNT-hybrid network-embedded PDMS film sensors can obtain highly reliable mechanical performance under continuous strain deformation, repeated stretch/release tests were conducted on the films. In any case, as shown in Figure 5 (I), the resistance of the SSF sensors sharply increases at the very first stretching and then returns to its initial value in the range of 15% to 25%. With the repetition of the test for ≥100 cycles under stretching conditions of 1%, 20% and 25%, the change in the resistance was also restored to its original position without any change in the resistance, as shown in Figure 5-(II).

Figure 2 shows the fabrication of the proposed highly sensitive strain sensor films based on the composite of functionalized PPy-NPs with the SWCNT–Ag NW hybrid networks embedded into the cross-linked PDMS elastomer surface layer owing to its high elasticity, flexibility, heat resistance and strong adhesion. After the peeling the cured PDMS from the Si wafer. Here, when liquid PDMS covers the SWCNT–Ag NW hybrid network layer, it penetrates into the interconnected pores of the two-dimensional (2D) SWCNT–Ag NW hybrid network because of its low viscosity and low surface energy. After curing, all 2D SWCNT–Ag NW hybrid networks are buried on the cross-linked PDMS surface without considerable voids, indicative of the successful transfer of SWCNT–Ag NW hybrid networks from Si wafers to PDMS. The Ag NW–SWCNT network layer comprising an Ag NW network interconnected with SWCNTs is embedded into the surface of ~50-µm-thick-PDMS films. Finally, a PPy–PA-complexed NP aqueous solution was directly coated on the upper surface of the Ag NW–SWCNT hybrid conductive layer, which exhibited optical transparency and good electrical conductivity. 

Before the gas detection test, the SSF sample was dried in a vacuum chamber at 80 °C for 30 min. The SSF sample was exposed to dry argon gas through the chamber for at least 10 min to stabilize the initial dc electrical resistance. Gas detection experiments were performed at 40 °C and the interference of humidity was also examined at 20, 50 and 80% RH conditions, respectively, as presented in Figure 6. Figure 6 (I) shows the flow of NH_3_ at 1 to 5 ppm for 30 s at 20% RH condition and the current resistance change at that time ad actual current resistance value. In addition, the flow of NH_3_ gas for 30 seconds was stopped and the chamber filled with dry argon gas until a steady current was reached. This process was repeated with varying concentrations of ammonia gas. Sensor response was evaluated as Δ*R*/*R*_0_ for ammonia, where *R_0_* (current resistance) was evaluated after argon-gas exposure and *R* was evaluated after exposure of NH_3_ gas. The normalized current resistance change, S (defined as S = 100 × Δ*R*/*R*_0_, where *R_0_* and *R* denote the current resistance in argon gas before exposure to NH_3_ and current resistance during exposure to NH_3_, respectively) was examined to measure the current variation. 

As can be seen from Figure 6 sensitivity reached 90% within a short response time (defined as the time to achieve a 90% current response) of 25 s. When film sensor was exposed to argon gas, the recovery time (defined as the time of a 90% current decrease) was ~700 s (Figure 6-(II)). However, as shown in the Figure 6-(I), the initial resistance (*R*_0_) before the introduction of 1 ppm NH_3_ gas was 6.31E + 02 ohm but their resistance value was rapidly increased to with the influx of ammonia. At this time, the change rate of the resistance value (S = 100×Δ*R*/*R*_0_) was 4.3% and current resistances (*R*) was 6.33E + 02 ohm. On the other hand, when NH_3_ gas was introduced at 2, 3, 4 and 5 ppm, respectively, under the 20% RH conditions, then S increased to 6.2, 8.2, 10.4 and 11.6%. Moreover, the base line is very stable after each cycle, indicating good real-time repeatability. In a steady-state response after 1500 s of exposure to gaseous ammonia as a function of ammonia concentration of 1 ppm to 5 ppm, the response shows a generally linear increase. However, as demonstrated by this work, this SSF provide very satisfactory performance results, especially when considered in terms of the response value under low temperature. In any case, the gas detection experiment in Figure 6 was performed on undrawn film and examined at 20% RH conditions. However, as shown in Figure 5, when the strain is stretched from 20% to 25%, the change value of the sheet resistance of the sensor film (initial: 30 Ω/sq) was increased to 55–75 Ω/sq, which is 1.8 to 2.5 times. When the stretched state was relaxed to the initial state, it was also found that the sheet resistance value of the stretched state was restored to the initial resistance. However, as shown in Figure 5, when the strain was stretched between 0% and 25%, even if the current resistance in the sensor film varies by 2.5 times and there was no significant difference in ammonia gas’s sensitivity to current change within this range. This means that the sensitivity to ammonia gas responsiveness does not differ significantly when the stretchable sensor dynamically expands and contracts. In addition, when the same procedure was performed at 50% RH and 80% RH with 1 ppm ammonia gas flow, then S was slightly increased to 5.8% and 5.7% but the influence of humidity was not so large, as indicated in Figure 6-(III). That is, it showed good humidity-resistant gas sensing characteristics. It seems to be because the sensing material (PPy-PA NP/Ag NW-SWCNT) is composed of a complex of conjugated carbon compounds with high electrical conductivity and the organic sensing group (PA) has a high binding force with ammonia gas. As a result, the change of the electrical resistance value with humidity was very small as 6% below. We experimented with argon, nitrogen and dry air conditions, respectively, to recover the sensor resistance to the original state. Similar results were obtained regardless of the type of inlet gas. However, in our experiment, by using dry argon gas, it was possible to keep the steady state current of the gas sensor more constant.

On the other hand, for comparison, reversible, reproducible responses of a sensor film comprising only PPy–PA NP with the injection of 3 ppm of NH_3_ were recorded as reference data (Figure 6-(II-b)) [21]. In this case, ~50% lower sensitivity was observed (Figure 6-(II-a)). However, the sensor films fabricated on the basis of the composite of functionalized PPy NP with SWCNTs–Ag NW hybrid networks were more sensitive to NH_3_ as compared with a sensor film comprising only PPy–PA-complexed NPs. This result is explained by the fact that SWCNTs–Ag NW hybrid networks as conductors extend the movement path of electrons and the electron movement becomes more active (the sensing mechanism of NH_3_, in the case of sensor films comprising PPy NP-complexed with PA, is explained in a previous paper [27]. Selectivity is another important parameter for a gas sensor in practical use. Our work further tested various volatile organic compounds (VOCs), such as acetone, formaldehyde, ethanol and toluene. The results were measured at the optimum operating conditions of each sensor to evaluate the selective properties. As clearly shown above, the PPY–PA NPs/Ag NW–SWCNT-hybrid network film sensor exhibits high selectivity for gaseous ammonia (S = 4 at 1 ppm) compared to other interfering analytes, to which it presents very high responses. On the other hand, in the case of acetone, formaldehyde, ethanol and toluene, the changes of resistance (S) with gas exposure were hardly observed at concentrations below 5 ppm, only when more than 100 ppm of gas was introduced could some amount of current change (S = ~4 at 100 ppm) be detected. These results suggest that the sensing material in this work can be confirmed to have a special selectivity in the ammonia gas composition. The experiments in this work are the result of sensitivity measurement under one gas condition and did not confirm the selectivity of specific gas component in two or more mixed gas environments.

## 4. Conclusions

In this study, stretchable gas sensor films based on a composite of functionalized PPy NPs with SWCNT–Ag NW hybrid networks embedded into the cross-linked PDMS elastomer with high sensitivity to NH_3_ were prepared. Using a small amount of 0.025 wt% of SWCNT to interconnect with Ag NWs, an Ag NW–SWCNT hybrid network conductor was obtained exhibiting excellent electrical properties of 30 Ω/sq. In particular, highly stretchable sensor films that can withstand strain were prepared using the Ag NW–SWCNT hybrid networks embedded into the PDMS layer. The sensor films exhibited a high elasticity of 25% and the strain sensor films exhibited a good response to the stretch/release of 100 cycles and hysteresis tests. In particular, these sensor films were highly sensitive to NH_3_ and exhibited improved reproducibility and reversibility upon exposure to NH_3_. These sensor films exhibited high reactivity of 1 ppm ammonia gas even at a low temperature of 40 °C with 20% RH and also maintained reproducibility. In addition, it was confirmed that the sensor film was hardly affected even at a humidity range of 20 to 80%. In particular, the stretchable strain gas sensing film in this work showed the potential to be used for detecting ammonia gas by attaching it to various wearable or portable film-sensor devices. 

## Figures and Tables

**Figure 1 nanomaterials-10-00696-f001:**
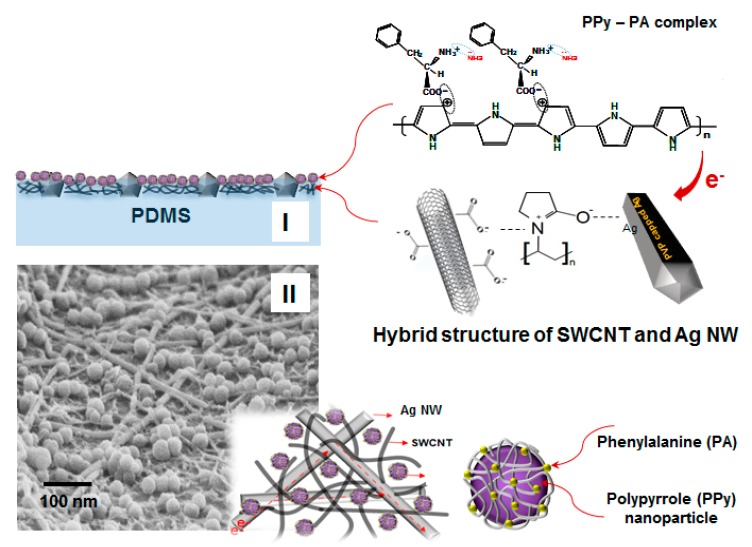
(**I**) Schematic of cross-sectional structure of stretchable sensor film (SSF), (**II**) Surface scanning electron microscope (SEM) image of SSF and schematic structures of PPy nanoparticle encapsulated in PA on SWCNT-Ag NW hybrid network. The figure above shows the chemical structure of PA combined with the conductive PPy backbone hybridized with single-walled carbon nanotube (SWCNT)-Ag NW hybrid network conductors.

**Figure 2 nanomaterials-10-00696-f002:**
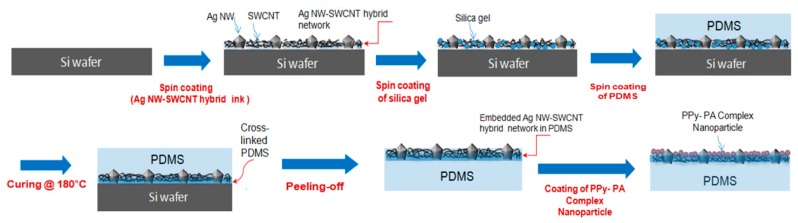
Fabrication of highly sensitive strain sensor films based on a nanocomposite of PPy-PA-complexed NPs with SWCNT–Ag NW hybrid networks embedded into the cross-linked polydimethylsiloxane (PDMS) elastomer.

**Figure 3 nanomaterials-10-00696-f003:**
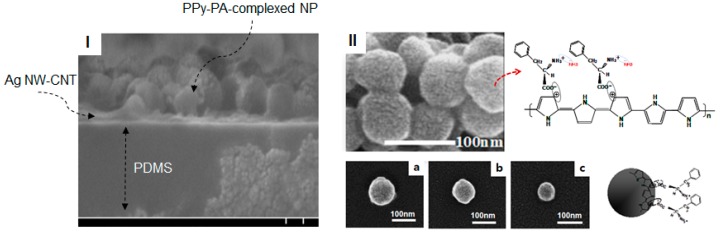
(**I**) Cross-sectional SEM image of SSF, (**II**) Schematic and chemical structures of PPy–PA-complexed NPs and their SEM images. At PVP molecular weights of 40,000, 360,000 and 1300,000, PPy-PA-complexed NPs of (**a**) 80–110 nm, (**b**) 60–90 nm and (**c**) 40–60 nm were synthesized, respectively.

**Figure 4 nanomaterials-10-00696-f004:**
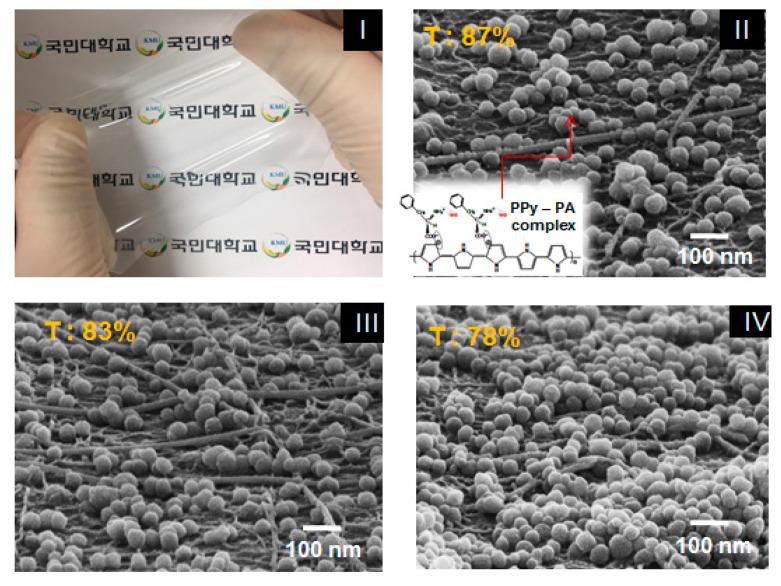
(**I**) Photograph of a film sensor sample and surface SEM images of the film sensor comprising a PPy–PA NP/Ag NW–SWCNT hybrid-network-embedded PDMS: (**II**) 1, (**III**) 2 and (**IV**) 3 wt% PPy–PA composite NP coated on a 30 Ω/sq Ag NW–SWCNT hybrid network electrode layer.

**Figure 5 nanomaterials-10-00696-f005:**
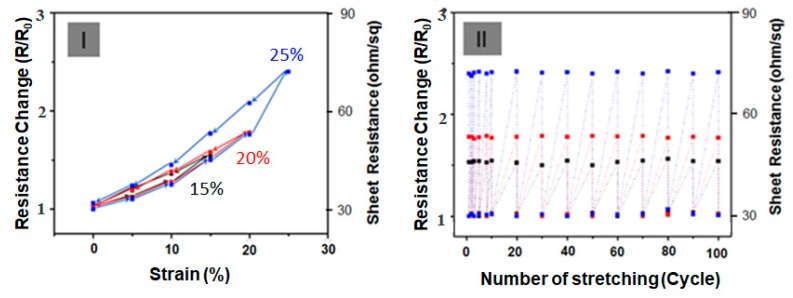
(**I**) Hysteresis curve of the film sensor comprising a PPy–PA NP/Ag NW–SWCNT-hybrid-network-embedded PDMS film (at tensile strains (ε) of 15%, 20% and 25%). (**II**) Effect of repeated stretching on the resistance change (R/R_0_) at strain recovery (stretch/release cycles).

**Figure 6 nanomaterials-10-00696-f006:**
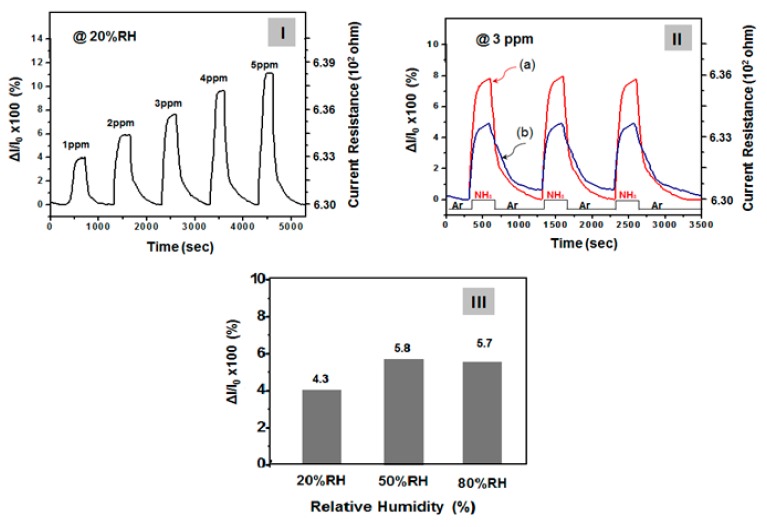
(**I**) Response of the film sensor based on PPy–PA NP/Ag NW–SWCNT hybrid-composites as function of ammonia concentrations ranging from 1 ppm to 5 ppm. (**II**-a) Reversible and reproducible responses of the film sensor comprising a PPy–PA NP/Ag NW–SWCNT hybrid-network-embedded PDMS film observed by the injection of 3 ppm of NH_3_ gas and the change of resistance during this period was recorded. Performance of the film sensor comprising only PPy–PA NP is shown in (**II**-b) for comparison [27]. (**III**) Sensor performance comparison under various humidity conditions (Reactivity of 1 ppm ammonia gas at a temperature of 40 °C).

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
