# Peer review of "Stretchable and High-performance Sensor films Based on Nanocomposite of Polypyrrole/SWCNT/Silver Nanowire"

_nanomaterials, 2020, doi:10.3390/nano10040696_

Round 1
Reviewer 1 Report
The authors propose a stretchable sensor films for sensors. Results for detection for ammonia, humidity and strech are presented. The reference list are complete and present recente works. The manuscript it is well written but the presentation of results need some improve.
Specific commnets:
- In the introduction refers a highly sensitive, conductive, extensible, and reliable “smart SSF” but don’t show in the manuscript. One comparation with similar is necessary
- I suggest change the title for section 2.
- Use SI units.
- The section 2.3 needs a clarification.
- Change the number for section “Results and discussion”
- The figure 1, 2 and 3 can be change for previous section. In this section only results.
- Is possible use this in optical fibers?
- Efect of temperature and the humidity?
- Is possible create a sensor for ammonia using this?
- The results in figure 6 is interesting but in the manuscript is confuse. Is necessary rewrite this.
- One conclusion is necessary.
I recommend the author rewrite the manuscript and present new version. A major review is necessary.
Reviewer 2 Report
The manuscript is dealing with fabrication and testing stretchable NH3 gas sensors based on nanocomposite thin film of polypyrrole-SWCNTs-Ag nanowires. This thin films exhibit outstanding mechanical elasticity (up to 25%) and relatively low electrical resistivity 30 Ohm/sq, as well as reproducible and reversible response to NH3. The composite was deposited by wet approach using spin-coating.
The topic of the manuscript is fitting to the journal scope and it is potentially interesting for a specific reader community. I recommend to improve text and add some results (see below mentioned experiments) prior the manuscript can be accepted for publication in the journal.
I have following questions and suggestions:
1/ Authors fabricated thin films of composite PPy-PA_NP/ Ag_NW-SWCNT and, importantly, compared its sensing response to 3ppm of NH3 (in dry air) to the response of film composed from PPy-PA_NP (in Fig.6 II). Authors did not compare these responses to PPy-PA_NP/ Ag film and to PPy-PA_NP/ SWCNT film. Such comparisons would help to understand the role of Ag_NW and SWCNT in the composite and discuss the synergy effect of their mixture in the presented composite type. Authors should add results of these experiments to Fig 6 II and discuss them.
2/ Authors did not provide information about SWCNTs. The different length of used SWCNTs can significantly influence electrical conductivity of the film at their same weight concentration in the film. Authors should add this SWCNT characteristic. It would be of reader interest to know diameter distributions of SWCNTs and G/D Raman intensity ratios of original SWCNTs, nanotubes after hydrothermal treatment, and after their incorporation into the composite.
3/ Authors should add the response curves of their sensor for dry air and air with 20% RH to Figure 5 I. (They should use the same measurement conditions, dry air/Ar; and humid air/Ar as in the presented sensor response).
4/ Authors should show results for sensor recovery in dry air after sensor response to NH3/air mixture and compare it to recovery in Ar.
5/ Authors should explain why they perform experiments at 40°C and not at room temperature and show the sensor response to NH3 for both cases.
Additionally I suggest to perform following changes in the text (or explain why they are used as they are used). (R.: is for referee recommendation)
In abstract: Authors write: “The SSF exhibited high conductivity of 30 ohm/sq …”???
R.: The property with the unit Ohm/sq is called resistivity not conductivity. Change it.
A/ In Abstract: line 19. “These SSF exhibited high reactivity…”
R.: It would be probably better to use word(s) (measurable) response to 1ppm as the words “high reactivity“
B/ In Abstract: line 21. R.: Authors should add word relative (humidity)
C/ In Introduction: line31. “… that respond to environmental or external stimuli…”
R.: Authors should rewrite this sentence because “environmental stimuli” are of external origin too.
D/ In Introduction: line 46. R.: I suggest to add word “electrical” tprior the word conductor.
E/ In 2.3 Sensor characterization and detection line131. R.: Please add type of used humidity sensor.
F/ R.: Change Nr 2 to Nr 3 in line 133 (3. Results…)
G/ line 190: Authors used term “excellent adhesion between SWCNT-Ag_NW and PDMS” without any reference or characterization proof or description.
R.: Add explanation for this statement.
H/ page 7 lines 236-240. Figure 5, R.: Increase size of the left y-axes labeling numbers to the same size as the label size on the right y axes. Identify curves (colors, symbols) for different values of strain.
I/ page 8 line 274. R.: Symbol for the electrical resistance used by authors, – I – , is very untypical. Resistance is usually symbolized by ro (Greek letter for r). Consider to change it.
J/ line 274: R.: Add equation for definition of the symbol S.
L/ lines 312-313. “… the change of the electrical resistance value with humidity is CONSIDER to be small.”
R.: Please proof it in respect to dry condition too . Add the results for dry condition to the figure 6. III.
M/ line 327: “sensor exhibit high selectivity for NH3 (S=4 at 1 ppm) compared to other interfering analytes”
R.: Prove it. Show please the sensor response to some other interfering analytes.
N/ line 338: R.: “4. Discussion” should be probably 4. Conclusions
Figure 6.: R.: Fig 6. I. is not showing reproducible response! It is not the reproducibility test. Change the text.
Reviewer 3 Report
I would recommend the submitted paper for publication
Author Response
Reviewer' letter checked
Round 2
Reviewer 1 Report
the authors present a reviewed version of previous manuscript where include change in response to previous comments. the change clarify the manuscript.
I suggest publish in present form
Author Response
Thank you gor reviewing our manuscript.
Reviewer 2 Report
The second version of the manuscript was partially improve but some parts are still missing scientific rigor and accuracy. Authors did not responds to some referee requests and recommendations for changes. It is pity that authors do not like to increase the scientific value of the manuscript adding additional experimental results but they can minimally improve the incorrect parts.
I will repeat some of my original comments/requests:
1/ In abstract:
Authors: “The SSF exhibited high conductivity of 30 Ω/sq and…”
R.: The property with the unit Ohm/sq is called electrical resistivity not conductivity. Electrical conductivity will be 1/(30 Ω/sq ). Of course, it can be use S =1/ Ω
R.: Change the sentence.
2/ R: Authors still did not provide information about SWCNTs. The different length of used SWCNTs can significantly influence electrical conductivity of the film at their same weight concentration in the film. Authors should add this SWCNT characteristic. (Each production charge even for the same producer could have different specifications!)
Reference 33 at line 86 is misleading
R.: Add SWCNTs specifications and add correct reference.
3/ Figure 6.: R.: Fig 6. I. is not showing reproducible response! It is not the reproducibility test. Change the text. The reproducible response is shown in Fig 6. II. Text in Figure caption is misleading!
R.; Change the text.
4/ Authors should explain why they perform experiments at 40°C and not at room temperature and show the sensor response to NH3 for both cases.
5/ In 2.3 Sensor characterization and detection, line143-144.
Authors: … “the humidity measurement was monitored using a high-performance humidity sensor installed in the gas flow path
R.: Please add type of used humidity sensor if you claim its “high-performance”.
6/ line 290: Authors ”However, in our experiment, when dry argon gas was used, it was easier to keep the steady state current of the gas sensor more constant.”
R: it is not possible to have something more constant as a constant…Rewrite the sentence.
